# TRAINING TRANSITION POLICIES VIA DISTRIBUTION MATCHING FOR COMPLEX TASKS

**Ju-Seung Byun & Andrew Perrault**
Department of Computer Science & Engineering
The Ohio State University
Columbus, OH 43210, USA
`{byun.83,perrault.17}@osu.edu`

## ABSTRACT

Humans decompose novel complex tasks into simpler ones to exploit previously learned skills. Analogously, hierarchical reinforcement learning seeks to leverage lower-level policies for simple tasks to solve complex ones. However, because each lower-level policy induces a different distribution of states, transitioning from one lower-level policy to another may fail due to an unexpected starting state. We introduce transition policies that smoothly connect lower-level policies by producing a distribution of states and actions that matches what is expected by the next policy. Training transition policies is challenging because the natural reward signal—whether the next policy can execute its subtask successfully—is sparse. By training transition policies via adversarial inverse reinforcement learning to match the distribution of expected states and actions, we avoid relying on task-based reward. To further improve performance, we use deep Q-learning with a binary action space to determine when to switch from a transition policy to the next pre-trained policy, using the success or failure of the next subtask as the reward. Although the reward is still sparse, the problem is less severe due to the simple binary action space. We demonstrate our method on continuous bipedal locomotion and arm manipulation tasks that require diverse skills. We show that it smoothly connects the lower-level policies, achieving higher success rates than previous methods that search for successful trajectories based on a reward function, but do not match the state distribution.

## 1 INTRODUCTION

While Reinforcement Learning (RL) has made significant improvements for a wide variety of continuous tasks such as locomotion (Lillicrap et al., 2016; Heess et al., 2017) and robotic manipulation (Ghosh et al., 2018), direct end-to-end training on tasks that require complex behaviors often fails. When faced with a complex task, humans may solve simpler subtasks first and combine them. Analogously, hierarchical reinforcement learning (HRL) uses multiple levels of policies, where the actions of a higher-level *meta-controller* represent which lower-level policy to execute (Sutton et al., 1999; Dayan & Hinton, 1992). A multitude of methods has been proposed to train HRL, such as assigning subgoals for the lower-level policies (Kulkarni et al., 2016), employing pre-trained lower-level policies (Frans et al., 2018), and off-policy RL (Nachum et al., 2018).

All HRL methods face a common challenge: switching smoothly from one lower-level policy to another when the meta-controller directs that a switch should occur. For example, running hurdles may require switching between running and jumping repeatedly while maintaining forward momentum. Previous approaches to this challenge (Frans et al., 2018; Andreas et al., 2017; Lee et al., 2019) involve either retraining existing lower-level policies and/or introducing a new policy to execute the switch. We focus on the latter and call the new policy the *transition policy*, following Lee et al. (2019).

A transition policy needs to switch between the states produced by one lower-level policy to an appropriate starting state for another. A natural way to train such a policy is to give a reward depending on whether the transition is successfully executed or not. However, this method induces a

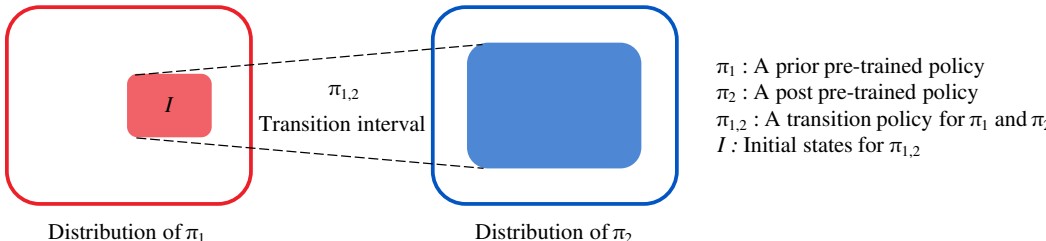

Figure 1: Transitions between lower-level policies for different subtasks often fail because one policy $\pi_2$ is not trained on states generated by another policy $\pi_1$. A transition policy $\pi_{1,2}$ starts from the state distribution induced by $\pi_1$ at time of transition. We propose training $\pi_{1,2}$ to produce states and actions from the distribution of $\pi_2$ using inverse reinforcement learning techniques.

sparse reward that is difficult to learn from due to the temporal credit assignment problem (Sutton, 1984). Instead, we use inverse reinforcement learning (IRL) techniques to match the distribution of states produced by the transition policy to the distribution of states expected by the next lower-level policy (Figure 1). While the goal of IRL is infer an expert's reward function from demonstrations of optimal behavior (Ng et al., 2000; Ziebart et al., 2008), some recent approaches to IRL (e.g., Finn et al. (2016); Ho & Ermon (2016); Fu et al. (2018)) learn a policy that imitates the expert's demonstrations in the process. In particular, Ho & Ermon (2016) propose a generative adversarial network (GAN) structure (Goodfellow et al., 2014) that matches the distribution of the policy (generator) with the distribution of the given data (expert demonstrations). We use this GAN structure for distribution matching, and, consequently, we avoid the problem of designing an explicit reward function for transition policies.

Although the transition policy is trained to produce the distribution of states and actions that is expected by the next lower-level policy, not all states in the distribution's support have equal probability of success. To increase the rate of successful execution, we introduce a deep Q-network (DQN) (Mnih et al., 2013) to govern the switch from the transition policy to the next lower-level policy. The DQN has two actions, *switch* or *stay*, and a simple reward function where a positive reward is obtained if the next lower-level policy is successfully executed; otherwise, a negative reward is given. The issue of sparse rewards is present here, but it is less of an issue because of the simple decision space.

The main contribution of this paper is a new method for training transition policies for transitioning between lower-level policies. The two parts of our approach, the transition policy that matches the distribution of states and the DQN that determines when to transfer control, are both essential. Without the DQN, the state selected for transition may be familiar to the next policy, but lead to a poor success rate. Without the transition policy, there may be no appropriate state for the DQN to transition from. The proposed method reduces issues with sparse and delayed rewards that are encountered by prior approaches. We demonstrate our method on bipedal locomotion and arm manipulation tasks created by Lee et al. (2019), in which the agent requires to have diverse skills and appropriately utilize them for each of the tasks.

## 2 PRELIMINARIES

The problem is formulated as a infinite-horizon Markov decision process (MDP) (Puterman, 2014; Sutton & Barto, 2018) defined by the tuple $\mathcal{M} = (\mathcal{S}, \mathcal{A}, \mathcal{P}, \mathcal{R}, \gamma, \mu)$. The agent takes action $a_t \in \mathcal{A}$ for the current state $s_t \in \mathcal{S}$ at time $t$, then the reward function $\mathcal{R} : \mathcal{S} \times \mathcal{A} \to \mathbb{R}$ returns the reward for $(s_t, a_t)$, and the next state $s_{t+1}$ is determined by the transition probability $\mathcal{P}(s_{t+1}|s_t, a_t)$. $\gamma$ is the discount factor and $\mu$ denotes the initial states distribution. The agent follows a policy $\pi_\theta(a_t|s_t)$ that produces a distribution of actions for each state. The objective of reinforcement learning (RL) is to find the optimal $\theta$ that maximizes the expected discounted reward:

$$\theta^* = \underset{\theta}{\operatorname{argmax}} \underset{\substack{s_0 \sim \mu \\ a_t \sim \pi_\theta(\cdot|s_t) \\ s_{t+1} \sim \mathcal{P}(\cdot|s_t, a_t)}}{\mathbb{E}} \left[ \sum_{t=0}^{\infty} \gamma^t R(s_t, a_t) \right]. \tag{1}$$

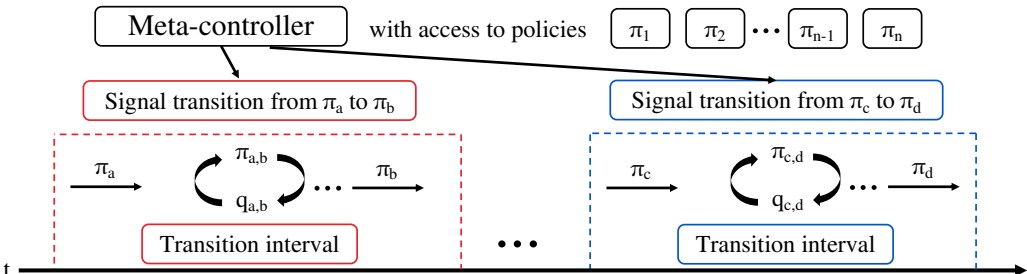

Figure 2: HRL structure with transition policies and DQNs. The meta-controller has $n$ pre-trained lower-level policies $\{\pi_1, \pi_2, ..., \pi_n\}$. To transition between two policies (for example, $\pi_a$ and $\pi_b$), the meta-controller signals a transition interval. We train a transition policy ($\pi_{a,b}$) that activates at the beginning of this interval and a DQN ($q_{a,b}$) that determines when during the interval the switch from $\pi_{a,b}$ to $\pi_b$ should take place.

## 2.1 INVERSE REINFORCEMENT LEARNING

Whereas standard RL trains a policy to maximize a reward function, inverse reinforcement learning (IRL) (Ng et al., 2000; Ziebart et al., 2008) seeks to learn the reward function from expert demonstrations of an optimal policy. Several approaches to the IRL problem (Ho & Ermon, 2016; Fu et al., 2018) adopt a GAN structure (Goodfellow et al., 2014)—the policy (generator) tries to select actions that the discriminator cannot distinguish from the expert's. As a by-product, the generator produces a policy with the same distribution as the expert's at convergence. We use the state-action discriminator (Fu et al., 2018) rather than the trajectory-centric discriminator (Finn et al., 2016; Ho & Ermon, 2016):

$$D_{\psi,\phi}(s_t, a_t, s_{t+1}) = \frac{\exp\{f_{\psi,\phi}(s_t, a_t, s_{t+1})\}}{\exp\{f_{\psi,\phi}(s_t, a_t, s_{t+1})\} + \pi_\theta(a_t|s_t)}, \tag{2}$$

where $f_{\psi,\phi}(s_t, a_t, s_{t+1}) = g_\psi(s_t, a_t) + \gamma h_\phi(s_{t+1}) - h_\phi(s_t)$ at time $t$, $g_\psi$ is the reward approximator, and $\phi$ are the parameters of a shaping term $h_\phi$, which alleviates unwanted shaping for $g_\psi$.

This discriminator $D_{\psi,\phi}$ is trained to distinguish data from the expert and the generator's policy via binary logistic regression, and the policy $\pi_\theta$ is updated with Proximal Policy Optimization (PPO) (Schulman et al., 2017) according to the rewards $r_{\psi,\phi}$ provided by the discriminator:

$$r_{\psi,\phi}(s_t, a_t, s_{t+1}) = \log D_{\psi,\phi}(s_t, a_t, s_{t+1}) - \log(1 - D_{\psi,\phi}(s_t, a_t, s_{t+1})) \tag{3}$$

## 2.2 DEEP Q-LEARNING

Deep Q-learning (Riedmiller, 2005; Mnih et al., 2013) is a variant of Q-learning (Watkins & Dayan, 1992) that produces a neural network called a deep Q-network (DQN) that aims to learn a deterministic optimal action from among a discrete set of actions for each state. Deep Q-learning suffers from overestimating $y_t$ (Thrun & Schwartz, 1993) because of the max term, which impedes the learning process. Double Q-learning (van Hasselt et al., 2016) addresses this problem by decoupling the networks for choosing the action and evaluating value. The target network $Q_{\theta'}$ makes learning more stable by providing a target $y_t$ that is fixed for long periods of time:

$$y_t' = r_t + \gamma \, Q_{\theta'}(s_{t+1}, \mathrm{argmax}_{a_{t+1}} Q_\theta(s_{t+1}, a_{t+1})) \tag{4}$$

The DQN is updated using a loss function such as the mean squared error $(y_t' - Q_\theta(s_t, a_t))^2$. We use train a DQN with double Q-learning to get a signal that tells us when to switch from a transition policy to the next lower-level policy.

## 3 APPROACH

In this section, we present a methodology for transitioning between lower-level policies in hierarchical reinforcement learning (HRL). We assume that we receive pre-trained lower-level subtask

---

**Algorithm 1** Training Transition Policy $\pi_{a,b}$

---

1: **Input:** pre-trained policies $\{\pi_a, \pi_b\}$, transition interval
2: Initialize transition policy (generator) $\pi_{a,b}$ and discriminator $D_{\psi,\phi}$ Equation (2)
3: Collect the states $\mathcal{S}_{\pi_a}$ of $\pi_a$ at the start of the transition interval
4: Collect the trajectories (states and actions) $\mathcal{T}_{\pi_b}$ of $\pi_b$ during the transition interval
5: Initialize transition policy $\pi_{a,b}$ and discriminator $D_{\psi,\phi}$
6: **for** $i = 1$ to $n$ **do**
7:     **for** $j = 1$ to $m$ **do**
8:         Sample $s \sim \mathcal{S}_{\pi_a}$ and set the state of $\pi_{a,b}$ to $s$
9:         Collect the trajectory $\mathcal{T}_{\pi_{a,b}}$ generated by the current $\pi_{a,b}$
10:     **end for**
11:     Train $D_{\psi,\phi}$ with binary logistic regression to distinguish $\mathcal{T}_{\pi_{a,b}}$ and $\mathcal{T}_{\pi_b}$
12:     Calculate reward $r_{\psi,\phi}$ set for all $(s_t, a_t, s_{t+1}) \in \mathcal{T}_{\pi_{a,b}}$ with Equation (3)
13:     Optimize $\pi_{a,b}$ with respect to $\mathcal{T}_{\pi_{a,b}}$ and $r_{\psi,\phi}$
14: **end for**

---

policies as input along with *transition intervals*—windows where a transition between two specific lower-level policies should occur. We seek to maximize performance on a complex task by combining the pre-trained policies without re-training them. To do this, we train a transition policy for each pair of pre-trained policies that we must transition between. The transition policy is activated at the start of each transition interval. Then, a DQN governs the switch from the transition policy to the next prescribed pre-trained policy. The DQN improves the rate of transition success relative to simply switching at the end of the transition interval.[1]

In Section 3.1, we describe our training procedure for transition policies. In Section 3.2, we describe how we collect data and design rewards for DQNs. Our framework is summarized by Figure 2.

## 3.1 TRAINING THE TRANSITION POLICY

Let $\{\pi_1, \pi_2, ..., \pi_n\}$ denote $n$ pre-trained policies and let $\pi_{a,b}$ denote a transition policy that aims to connect policy $\pi_a$ to $\pi_b$. The meta-controller dictates a transition interval for each transition between pre-trained policies—a period of time when the transition should take place. If we were to simply switch from $\pi_a$ to $\pi_b$ at the start of this interval, $\pi_b$ can easily fail because it has never been trained on the last state produced by $\pi_a$. To prevent this problem, $\pi_{a,b}$ needs to be able to start from any of the last states produced by $\pi_a$ and lead to a state from which $\pi_b$ can successfully perform the next task. Which states are favorable for starting $\pi_b$? A possible answer is the states produced by $\pi_b$ itself.

We train $\pi_{a,b}$ to have the same distribution as $\pi_b$ with IRL (Ng et al., 2000; Ziebart et al., 2008; Ho & Ermon, 2016; Fu et al., 2018). We specifically choose adversarial inverse reinforcement learning (AIRL) (Fu et al., 2018), which has the GAN (Goodfellow et al., 2014) structure. The training procedure for $\pi_{a,b}$ is described in Algorithm 1. $\pi_{a,b}$ is trained with two classes of data: one from $\pi_a$ and one from $\pi_b$. From $\pi_a$, we collect the states $\mathcal{S}_{\pi_a}$ of $\pi_a$ at the start of the transition interval. From $\pi_b$, we collect the trajectories $\mathcal{T}_{\pi_b}$ of $\pi_b$ during the the transition interval. The initial state of the generator $\pi_{a,b}$ is set to $s$, sampled from $\mathcal{S}_{\pi_a}$. Then, $\pi_{a,b}$ tries to generate trajectories that fool the discriminator $D$ (Equation 2) into classifying it as an example of $\mathcal{T}_{\pi_b}$. The discriminator $D$ is trained with binary logistic regression to distinguish between the data from $\pi_b$ vs. $\pi_{a,b}$. When $D$ can no longer distinguish between them, $\pi_{a,b}$ can be said to have a similar distribution to $\pi_b$.

For example, we prepared a pre-trained *Walking forward* policy $\pi_1$ and *Jumping* policy $\pi_2$ for one of our experimental environments, *Hurdle*, manually shaping rewards for these two subtask policies. We use a transition interval of 3–4 meters from the hurdle. Thus, $\mathcal{S}_{\pi_1}$ is the last states of $\pi_1$—the sampled set of states when the agent's location is 4 meters from the hurdle. $\mathcal{T}_{\pi_2}$ is the trajectories

---

[1]We assume that the environment is static enough that it is not critical that the meta-controller has the opportunity to replan after the transition policy terminates. In principle, such replanning could be allowed by introducing an additional layer of transition policies, i.e., by training training meta-transition policies that transition between transition policies and actions.

---

**Algorithm 2** Training Deep Q Network $q_{a,b}$

---
1: **Input:** pre-trained policies $\{\pi_a, \pi_b\}$, transition policy $\pi_{a,b}$, transition interval
2: Initialize $q_{a,b}$ and replay buffer $B$
3: **for** $i = 1$ to $n$ **do**
4:     Set environment to state of $\pi_a$ at the start of the transition interval
5:     **while** the current state is in the transition interval
6:         $a_t \sim \pi_{a,b}(s_t)$
7:         Execute environment for $(s_t, a_t)$ and get $s_{t+1}$ and success-fail-alive signal $\tau$
8:         Run $q_{a,b}$ and get an action $a_q$
9:         **if** $a_q = switch$
10:             set $s = s_{t+1}$ and break
11:         **else if** $a_q = stay$
12:             Store $(s_t, a_q, r_f, s_{t+1})$ in $B$ if $\tau = $ fail or $s_{t+1}$ is outside the transition interval
13:             Store $(s_t, a_q, 0, s_{t+1})$ in $B$ if $\tau = $ alive
14:         **end if**
15:     **end while**
16:     **while** $\pi_b$ has not achieved subgoal and has not failed
17:         $a_t \sim \pi_b(s_t)$
18:         Execute environment for $(s_t, a_t)$ and get $s_{t+1}$ and signal $\tau$
19:     **end while**
20:     Store $(s, a_q, r_s, s_{t+1})$ in $B$ if $\tau = $ success
21:     Store $(s, a_q, r_f, s_{t+1})$ in $B$ if $\tau = $ fail
22:     Update $q_{a,b}$ with a minibatch from $B$ according to Equation (4)
23: **end for**

---

of $\pi_2$ from 3–4 meters from the hurdle. The transition policy $\pi_{1,2}$ is initialized with one of the last states of $\pi_1$ and learns to match the distribution $\pi_2$ just before the jump is initiated.

## 3.2 TRAINING THE DEEP Q-NETWORK FOR TRANSITION

With $\pi_{a,b}$, we can proceed from the last states of the $\pi_a$ to the distribution of states of $\pi_b$. In order to select a state to switch from $\pi_{a,b}$ to $\pi_b$ that is more likely to lead to a successful execution of $\pi_b$, we introduce a DQN (Riedmiller, 2005; Mnih et al., 2013; van Hasselt et al., 2016), which we denote as $q_{a,b}$. At each time step, $q_{a,b}$ outputs whether to *switch*, i.e., hand control from $\pi_{a,b}$ to $\pi_b$, or *stay*, i.e., continue to use $\pi_{a,b}$.

As shown in Figure 3, we design a simple reward function to train $q_{a,b}$. The process begins when the agent enters the transition interval and switches from $\pi_a$ to $\pi_{a,b}$ at one of the states in $\mathcal{S}_{\pi_a}$. Then, $q_{a,b}$ becomes active and generates a *switch* or *stay* action at each subsequent time step, according to the current state. If $q_{a,b}$ takes the *switch* action, the agent switches to $\pi_b$. The reward for this transition depends on the outcome of the subtask that $\pi_b$ aims to complete. A positive value $r_s$ is given if the agent completes the next subtask; otherwise, the agent gets a negative reward $r_f$. If $q_{a,b}$ takes the *stay* action, the policy $\pi_{a,b}$ is continued. The agent immediately receives an alive reward of 0 unless the agent enters a failure state, in which case it receives a reward of $r_f$. (For example, in the *Hurdle* environment, failure states are those where the agent collapses.)

$q_{a,b}$ is exposed to a sparse reward issue because of the delayed reward for the *switch* action, but the issue is less severe compared to training the entire transition policy using the sparse subtask success reward. The multi-dimensional continuous control space is reduced to a single dimensional binary decision.

In summary, we introduce a method to train transition policies that smoothly connects pre-trained policies without having to design a complex reward function and fine-tune the pre-trained policies. We use a DQN to increase the performance of the following pre-trained policy by controlling the timing of the switch from the transition policy to the next pre-trained policy.

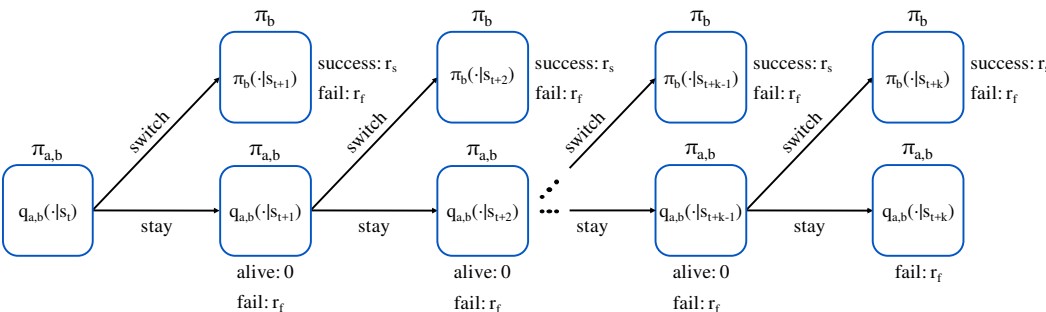

Figure 3: The sequential decision problem faced by the DQN during the transition interval. The transition interval runs from $s_t$, the ending state of $\pi_a$, to $s_{t+k}$. The DQN $q_{a,b}$ is active until it outputs a *switch* action, which passes control from the transition policy $\pi_{a,b}$ to $\pi_b$. The *switch* reward depends on whether $\pi_b$ achieves the subgoal: a reward of $r_s$ is given for success a reward of $r_f$ for failure. A reward of 0 is given for the *stay* action unless the agent enters a failure state, in which case $r_f$ is given.

## 4 RELATED WORK

Training agents that can generate diverse behaviors in complex environments is still an open problem in RL. Although some complex tasks can be solved by careful modification of the reward function (Ng et al., 1999), the modifications are not simple and may produce unexpected behaviors (Riedmiller et al., 2018). HRL approaches may be able to avoid these issues by combining lower-level policies for subtasks that are easier to train. We briefly summarize relevant past work in HRL.

The option framework (Sutton et al., 1999) use a two-level hierarchy: the higher-level policy is in charge of selecting the lower-level policies (options) and selecting the termination conditions for each option. The option-critic architecture for continuous problems (Bacon et al., 2017) trains the higher-level policy jointly with the options. This approach may lead to options that terminate at every step, flattening the hierarchy. Combining pre-trained policies has been shown to be successful for learning diverse behaviors, under the assumption that the subtasks are known and well-defined (Heess et al., 2016; Florensa et al., 2017). Given subgoals and pre-trained policies, the h-DQN endeavors to use the lower-level policies to achieve the subgoals (Kulkarni et al., 2016). Frans et al. (2018) proposed an approach to solving unseen tasks by using pre-trained policies with the master policy. The feudal reinforcement learning framework (Dayan & Hinton, 1992) creates a hierarchy where there are managers and sub-managers in which a manager learns to assign subgoals to their sub-managers (these sub-managers also have their sub-managers). All of these approaches require switching between lower-level policies, and, in principle, their performance could be improved by a transition policy. Because we do not alter the pre-trained policies at all, our method may perform poorly if the task is unseen and the pre-trained policies must be fine-tuned to perform well or if a successful transition *requires* that the pre-trained policies are altered.

Our contribution is most closely related to past work on how to transition between lower-level policies. Tidd et al. (2021) introduced estimators that decide when to switch the current policy to the next policy. They trained the estimators with the assumption that the policies have common states. Our work is inspired by past approaches that train transition policies to connect sub-policies smoothly. Lee et al. (2019) introduced a proximity predictor that predicts the proximity to the initial states for the next policy. The proximity predictor is used as a dense reward function for the transition policies, which otherwise struggle with sparse rewards. In our work, we approach the sparse reward issue with distribution matching, and instead of transition policies producing signals for switching policies (Lee et al., 2019), we have independent Q-networks that give the transition signals.

An alternative to using generative adversarial reinforcement learning to match the state and action distribution would be to imitate the trajectories directly with behavioral cloning (Bain & Sammut, 1995; Ross & Bagnell, 2010). However, behavioral cloning suffers from the distribution drift problem in which, the more the policy generates actions, the more the actions tend to drift away from the distribution of the expert; hence, we use the GAN structure of IRL for distribution matching.

Table 1: Success count table for three arm manipulation tasks. Each entry represents an average success count with a standard deviation over 50 simulations with three different random seeds. Our method performs better than using a single policy trained with TRPO and shows close results to Lee et al. (2019). Both Lee et al. (2019) and our method approach perfect success counts (5 for *Repetitive picking up*, 5 for *Repetitive catching*, and 1 for *Serve*).

|  | Repetitive picking up | Repetitive catching | Serve |
|---|---|---|---|
| Single | $0.69 \pm 0.46$ | $4.54 \pm 1.21$ | $0.32 \pm 0.47$ |
| Without TP | $1.49 \pm 0.59$ | $4.58 \pm 1.10$ | $0.05 \pm 0.22$ |
| With TP | $3.55 \pm 1.64$ | $4.75 \pm 0.87$ | $0.99 \pm 0.81$ |
| Lee et al. (2019) | $\mathbf{4.84 \pm 0.63}$ | $\mathbf{4.97 \pm 0.33}$ | $0.92 \pm 0.27$ |
| With TP and Q (Ours) | $4.77 \pm 0.91$ | $4.76 \pm 0.82$ | $\mathbf{1.00 \pm 0.00}$ |

## 5 EXPERIMENTS

We evaluate our method on six continuous bipedal locomotion and arm manipulation environments created by Lee et al. (2019): *Repetitive picking up*, *Repetitive catching*, *Serve*, *Patrol*, *Hurdle*, and *Obstacle course*. All of the environments are simulated with the MuJoCo physics engine (Todorov et al., 2012). The tasks are detailed in Section 5.1 and Section 5.2. Section 5.3 shows the results— our method performs better than using only a single policy or simply using pre-trained policies and is comparable to Lee et al. (2019) for the arm manipulation tasks and much stronger for the locomotion tasks. In Section 5.4, we visualize how a transition policy trained through IRL resembles the distribution of pre-trained policies.

### 5.1 ARM MANIPULATION TASKS

A Kinova Jaco arm with a 9 degrees of freedom and 3 fingers is simulated for three arm manipulation tasks. The agent needs to do several basic tasks with a box. The agent receives the robot and the box's state as an observation.

**Repetitive picking up:** *Repetitive picking up* requires the robotic arm to pick up the box 5 times. After picking up the box from the ground, the agent needs to hold the box in the air for 50 steps (to demonstrate stability). If the agent does so, the environment increments the success count by 1, and the box is randomly placed on the ground. We prepared a pre-trained policy *Pick* trained on an environment where the agent needs to pick up and hold the box once.

**Repetitive catching:** *Repetitive catching* requires the robotic arm to catch the box 5 times. After catching the box, the agent needs to hold the box in the air for 50 steps. If the agent does so, the environment increments the success count by 1, and the next box is thrown 50 steps later. The pre-trained policy *Catch* is trained on an environment that requires the agent to catch and hold the box once.

**Serve:** *Serve* is a task in which the agent picks up a box from the ground, tosses it, and then hits the box at a target. For this task, there are two pre-trained policies *Tossing* and *Hitting*. The performance without using a transition policy is very poor because *Hitting*'s initial states are different from the ending states of *Tossing*. By introducing the transition policy connecting *Tossing*'s end states to the start states of *Hitting* smoothly, the agent performs the task perfectly.

### 5.2 BIPEDAL LOCOMOTION TASKS

The agents for three locomotion tasks have the same configuration. The agent is a 2D bipedal walker with two legs and one torso and 9 degrees of freedom. The agent receives (joint information, velocity, distance from an object) tuples as the state. All of the locomotion tasks require the agents to do diverse behaviors to accomplish each subgoal.

Table 2: Success count table for three locomotion tasks. Each entry represents an average success count with a standard deviation over 50 simulations with three different random seeds. Our method performs better than using a single policy trained with TRPO and Lee et al. (2019).

|  | **Patrol** | **Hurdle** | **Obstacle Course** |
|---|---|---|---|
| Single | $1.37 \pm 0.52$ | $4.13 \pm 1.54$ | $0.98 \pm 1.09$ |
| Without TP | $1.26 \pm 0.44$ | $3.19 \pm 1.61$ | $0.41 \pm 0.52$ |
| With TP | $3.11 \pm 1.58$ | $3.43 \pm 1.60$ | $2.44 \pm 1.63$ |
| Lee et al. (2019) | $3.33 \pm 1.38$ | $3.14 \pm 1.69$ | $1.90 \pm 1.45$ |
| With TP and Q (Ours) | $\mathbf{3.97 \pm 1.38}$ | $\mathbf{4.84 \pm 0.60}$ | $\mathbf{3.72 \pm 1.51}$ |

**Patrol:** *Patrol* is a task in which the agent alternately touches two stones on the ground. For this task, the agent must walk forwards and backwards. We trained subtask policies *Walk forward*, *Walk backward*, and *Balancing*. *Balancing* helps alleviate the extreme change between *Walk forward* and *Walk backward*. The initial direction is determined randomly, and the corresponding pre-trained policy is executed. Before switching to the opposite direction's pre-trained policy, balancing runs for 100 steps.

**Hurdle:** In *Hurdle* environment, the agent needs to walk forward and jump over a curb 5 times. The two subtask policies are *Walking forward* and *Jumping*. Each of the subtask policies are trained on a sub-environment: *Walking forward* is trained to walk 1000 steps on a flat surface, and *Jumping* learns to jump over one obstacle. As shown in Table 2, our method works very well and is close to the perfect success count.

**Obstacle Course:** *Obstacle course* has two types of obstacles, curbs and ceilings. There are two curbs and three ceilings; hence the agent needs to be able to walk, jump, and crawl to get through the obstacles. The order of obstacles is randomly decided before starting. We prepared three subtask policies: *Walking forward*, *Jumping*, and *Crawling*. Each subtask policy was trained on the sub-environment of *Obstacle course* to complete each task.

## 5.3 COMPARATIVE EVALUATION

To prove that utilizing pre-trained policies is advantageous when faced with complex problems, we compare our method with the performance of a single end-to-end policy (Single), only using pre-trained policies without transition policies (Without TP), and with transition policies (With TP), and Lee et al. (2019)'s method. All subtask policies and the single policy are trained with TRPO (Schulman et al., 2015). Note that we also conducted tested other training algorithms such as PPO (Schulman et al., 2017) and SAC (Haarnoja et al., 2018), but the results were comparable to TRPO. Table 1 and 2 are the final results, showing that our method of using transition policies and DQNs has higher success rates than using a single policy. All arm manipulation tasks achieve close to perfect scores, similar to Lee et al. (2019), and in the locomotion tasks, our method outperforms all baselines.

## 5.4 VISUALIZING TRANSITION POLICY DISTRIBUTION

The goal of our transition policy is to start from the ending states of one pre-trained policy, and then induce a trajectory with the same distribution as another pre-trained policy during the transition interval, which train using IRL employing the GAN structure. We can see that the success rates significantly increase when this kind of transition policy is used (Tables 1 and 2), relative to a single policy trained end-to-end or pre-trained policies with immediate transitions. These results can be interpreted as an indication that our transition policy has succeeded in distribution matching. Figure 4 illustrates t-SNE results for the actions generated by *Walking forward→Balancing* and *Tossing→Hitting*. Green dots mean the distribution of the transition policy in the corresponding transition interval. They are located between the distributions of the other two colors, an indication that a transition policy is able to bridge two different pre-trained policies.

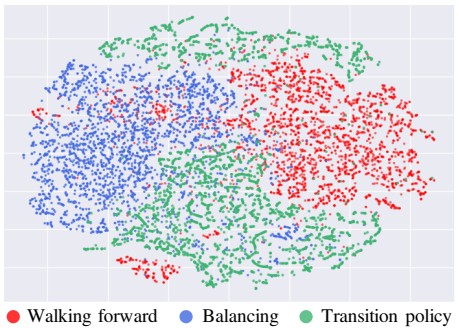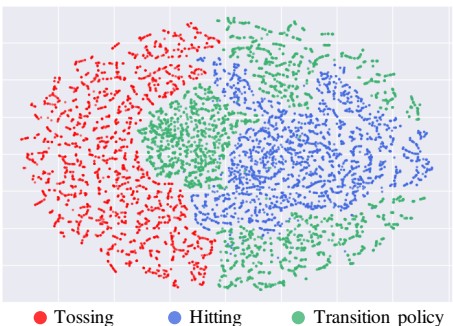

● Walking forward    ● Balancing    ● Transition policy          ● Tossing    ● Hitting    ● Transition policy

Figure 4: Three classes of distributions projected into 2D space with t-SNE. Both figures illustrate the actions of two pre-trained policies and the corresponding transition policy connecting them in a transition interval. The left figure shows for *Walking forward→Balancing*, and the right one stands for *Tossing→Hitting*. The actions from the transition policy lie between those of the pre-trained policies

## 6    CONCLUSION

We have presented an approach for HRL in which an agent utilizes pre-trained policies for simpler tasks to solve a complex task. Transitions between pre-trained policies are essential; hence, pre-trained policies are often fine-tuned to suit a complex task (Frans et al., 2018). Instead of fine-tuning, we introduce transition policies (Lee et al., 2019) to connect the pre-trained policies. Since our transition policy learns the partial distribution of the following pre-trained policy through IRL, we do not need to create a complicated reward function for a transition policy. A DQN is used to detect states with a high transition success rate as additional assistance. HRL frameworks often have trouble training the structure when rewards are sparse. We alleviate this issue by exposing only exposing a DQN with a binary action space to the sparse reward signal. The DQN is less susceptible to the sparse reward issue than the transition policy with continuous control. In our evaluation, we find the proposed method to be competitive with previous work in arm manipulation and much stronger in bipedal locomotion.

Our framework assumes that we can prepare suitable pre-trained subtask policies for HRL. This assumption is a strong requirement because many complex tasks are difficult to decompose into smaller subtasks, and even if decomposition is possible, the subtask policies may need to depend on the next subgoal, i.e., if an object needs to be picked up in a certain way to allow for success in placement. Nevertheless, since our transition policy training procedure is not dependent on exploration and sparse rewards, we believe that our work can give meaningful help to existing hierarchical methods and open new directions for HRL. Even in the case where subtask policies must be retrained to achieve success, the distribution matching approach may be useful, e.g., by mixing the subtask reward with the discriminator's reward.

### REPRODUCIBILITY STATEMENT

We describe the learning architectures, training procedures, as well as how we select transition intervals and hyperparameters in the appendix. We also prepared the pre-trained policies we use to facilitate follow-up work. The source code is available at `https://github.com/shashacks/IRL_Transition`.

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

## A  APPENDIX

To verify our method, we use six environments—three of them are robotic arm manipulation tasks and the other three are for the bipedal locomotion tasks. The arm manipulation tasks are conducted on a floor composed of square (0.1 m × 0.1 m) tiles. The agent receives an input size of 49 as an observation. An observation consists of the robotic arm's information like joints and velocity and the box's state. The bipedal locomotion tasks are conducted on a floor composed of square (1 m × 1 m) tiles and receive an input size of 26, plus an additional input when obstacles exist. The details are in Section B.2.

## B    PRE-TRAINED POLICY

All policies are trained wIth PPO and provided by Lee et al. (2019) for a fair comparison. The pre-trained policies have a two-layer *tanh* network with 32 units. For each epoch, we collect a rollout of size 10000 and use the Adam optimizer with mini-batch size of 64 and learning rate 1e-3 to update a policy.

### B.1    ROBOTIC ARM MANIPULATION

There are four pre-trained policies (*Picking, Catching, Tossing,* and *Hitting*). *Picking* is used for the task *Repetitive picking up* requiring the agent to pick the box five times, and *Catching* is used for the task *Repetitive catching* requiring the agent to catch the box five times. The task *Serve* utilizes *Tossing* and *Hitting* to serve like tennis.

***Picking***: The agent for *Picking* aims to pick the box one time on the floor. The box is randomly placed on a 0.1 m $\times$ 0.1 m tiles with a center (0.5, 0.2). After picking up the box, the agent needs to hold it 50 steps in the air.

***Catching***: The agent for *Catching* aims to catch the box one time. The box is thrown from the air to near the agent. After catching the box, the agent needs to hold it 50 steps in the air. The box is located at (0.0, 2.0, 1.5) and thrown for the next catching.

***Tossing***: The agent for *Tossing* aims to toss the box for serve. The box is randomly placed on a 0.005 m $\times$ 0.005 m tiles with a center (0.4, 0.3, 0.005).

***Hitting***: The agent for *Hitting* aims to hit the box and the box needs to go towards the given target. The box is randomly placed at position 0.005 m $\times$ 0.00 5m square region with a center (0.4, 0.3, 1.2).

### B.2    BIPEDAL LOCOMOTION

There are five pre-trained policies (*Walking forward, Walking backward, Balancing, Jumping* and *Crawling*). The task *Patrol* requires three pre-trained policies: *Walking forward, Walking backward,* and *Balancing* to move back and forth. The task *Hurdle* uses *Walking forward* and *Jumping* to pass five curbs. The task *Obstacle course* utilizes *Walking forward*, *Jumping*, and *Crawling* to pass curbs and ceilings.

The observation of *Patrol* has an additional observation of size 1. The additional observation is for a distance between the agent and the stone that needs to be touched. The *Obstacle course* task has two additional observations: one is for the distance between the agent and the starting point of an obstacle, and the other is for the distance between the agent and the obstacle ending point.

***Walking forward***: The agent for *Walking forward* must walk forward for 1000 steps. This agent receives size 26 input as an observation.

***Walking backward***: The agent for *Walking backward* must walk backward for 1000 steps. This agent receives size 26 input as an observation.

***Balancing***: *Balancing* requires the agent to balance in place for 1000 steps. This pre-trained policy is used to control extreme changes in *Walking forward* and *Walking backward*. This agent receives 26 size input as an observation.

***Jumping***: *Jumping* requires the agent to jump over a curb. The curb size is height 0.4 m and length 0.2 m and is randomly located between 5.5 m–8.5 m. This agent receives size 28 input as an observation.

***Crawling***: *Crawling* requires the agent to crawl under a ceiling. The ceiling size is height 1.0 m and length 16 m and is located on 2.0 m. This agent receives size 26 input as an observation.

Table 3: Update numbers for the arm manipulation tasks with how many interactions are required to get the success-fail data.

|  | Repetitive picking up | Repetitive catching | Serve |
|---|---|---|---|
| # Update | 15000 | 20000 | 25000 |
| # Interaction | 1700000 | 3700000 | 4100000 |

## C  TRANSITION POLICY

To get transition polices, we use IRL, specifically the state-action discriminator (Equation (1)) proposed in AIRL [8]. The generator (transition policy) has the same network size as the pre-trained policy and it is also trained with PPO according to rewards provided by the discriminator. The state-action discriminator $D_{\psi,\phi}(s_t, a_t, s_{t+1})$ contains the reward approximator $g_\psi(s_t, a_t)$ and the shaping term $h_\phi(s_t)$ that each of the networks has a two-layer *ReLU* network with 100 units.

We collect the last 10000 states that we define for the preceding pre-trained policy and one million trajectories for the subsequent pre-trained policy. To train a transition policy for our task, the transition policy interacts with the corresponding environment 10 million times, and every 50000 times, the transition policy and the corresponding discriminator are updated. We use the Adam optimizer (Kingma & Ba, 2015) with mini-batch size of 64 and learning rate 1e-4 for the transition policies and the learning rate 3e-4 for the discriminators.

***Repetitive picking up***: One transition policy, *Picking→Picking*, is used for this task. The transition interval is from after holding the box 50 steps to before picking the next box on the floor. We collect *Picking*'s demonstration of starting from the beginning to holding the box for 50 steps; hence the transition policy starts from the air and pick up the next box.

***Repetitive catching***: One transition policy, *Catching→Catching*, is used for this task. The transition interval is from after catching the box and wait for 50 step to before catching the next box. We collect *Catching*'s demonstration of starting from the beginning to until 85 steps.

***Serve***: One transition policy, *Tossing→Hitting*, is used for this task. The transition interval is from the box passes 0.7 m by *Tossing* to before hitting the box. We collect *Hitting*'s demonstration from after 20 steps to until hitting the box.

***Patrol***: There exist four transition policies, *Walking forward↔Balancing* and *Walking backward↔Balancing*. The transition intervals are the same except for the position. The transition intervals for *Walking→Balancing* is from when the agent pass a stepping stone to balance for 100 steps. The transition interval for *Balancing→Walking* is from after balancing 100 steps to until *Walking* walks 4 m to the corresponding direction.

***Hurdle***: There exist two transition policies, *Walking forward↔Jumping*. The transition interval of *Walking forward→Jumping* is 3 m–4 m from the curb. The transition interval of *Jumping→Walking forward* is 2.5 m–3.5 m from when the agent jumps over the curb. We collect the *Walking forward*'s demonstration from the beginning to 4 m.

***Obstacle course***: There exist four transition polices, *Walking forward↔Jumping* and *Walking forward↔Crawling*. The transition interval of *Walking forward→Jumping* is 2 m–3 m from the curb, and *Jumping→Walking forward* is the same as *Hurdle*. The transition interval of *Walking forward→Crawling* is 2 m–3 m in front of a ceiling, and *Crawling→Walking forward* is 2 m–3 m after passing the ceiling. *Walking forward→Crawling* uses the same *Walking forward* trajectories as *Walking forward→Jumping*.

## D  DEEP Q NETWORK

Our DQNs have a two-layer *ReLU* with 128 units. The Adam optimizer is used with learning rate 1e-4 and mini-batch size of 64 to update DQNs. All of the replay buffers of DQNs store one million samples. We simply set $r_s$ to 1 and $r_f$ to -1 for all of the six environments. If DQNs can be trained

Table 4: Update numbers for the bipedal locomotion tasks with how many interactions are required to get the success-fail data.

|             | **Patrol** | **Hurdle** | **Obstacle Course** |
|-------------|------------|------------|---------------------|
| # Update    | 30000      | 20000      | 30000               |
| # Interaction | 30000000 | 11000000   | 47000000            |

separately like when training transition policies, the total interaction with an environment can be greatly reduced. However we trained all DQNs together here and the number of interactions and updates can be seen in Table 3 and 4.

