# OpenReview forum: "Training Transition Policies via Distribution Matching for Complex Tasks"
_ICLR.cc/2022/Conference — ICLR 2022 Poster_

### Official Review · Reviewer_1ZCq · 2021-11-02

**Correctness:** 4
**Technical Novelty And Significance:** 3
**Empirical Novelty And Significance:** 3
**Recommendation:** 6
**Confidence:** 2

**Main Review:**

Let me start with the caveat that I am not an expert on RL, so that my knowledge of related work is limited. This said, I found the paper remarkably well-written and easy to follow. I have only a few questions below, some of which may be due to lack of expertise.

* You characterise your technique of policy distribution matching as an instance of Inverse RL, but in the first paragraph of p.2, you say that the technique "avoid[s] the problem of designing an explicit reward function ...", which looks to be conflicting with the term "Inverse RL". Can you please clarify ?

* In Tables 1 and 2, while the "Single" and "With TP" rows are reasonably clear from the text, unless I missed it, I could not find a description of "Without TP".

* You do not say much on how you determine the success-fail rewards $r_s,r_f$ hyperparameters. Can you clarify?

* At a more general level, a number of important aspects seem to be left to manual decisions, such as the design of the subtasks, their order, or the definition of the transition intervals. Should the non-expert reader assume that the automation of such decisions is beyond the current state of art?

**Summary Of The Paper:**

The paper considers the problem of solving a complex task requiring different skills by combining "subtask" policies pretrained for each individual skill. One approach here consists in finding ways to transition smoothly between the policy $b$ for a subtask and the policy $b$ for the next subtask. The paper proposes to do that based on training a "transition policy" that starts from a state produced by $a$ and attempts to match the _distribution_ of state-action pairs associated with $b$. Additionally, a DQN controls the exact timing at which the transition policy passes the control to $b$, which improves the success rate of the transition. Experiments are conducted on some simulated arm manipulation and bipedal locomotion tasks with similar results to current baselines on the first group and superior results on the second group.


**Summary Of The Review:**

From the position of a non-expert of the domain, I found the paper well-written and instructive, and was convinced by the idea of matching the transition policy and target policy distributions, but cannot really judge the novelty.

---

> ### Author Response · Authors · 2021-11-14
> **Author response for reviewer 1ZCq**
>
> Thank you for your review.
>
> (R1) You characterise your technique ... Can you please clarify?\
> => Our point is that the inverse RL techniques, while they may learn a reward function in the process of distribution matching, do not require us to design a reward function for the transition policy. Past work (Lee et al.) has approached the problem by attempting to design a reward function that the transition policy can be trained with—our approach avoids this explicit design step by using an imitative objective to train the transition policy.
> \
> &nbsp;
>
> (R2) In Tables 1 and 2 ... could not find a description of "Without TP". \
> => We have added a description of Without TP to Sec 5.3.
> \
> &nbsp;
>
> (R3) You do not say much on ...  Can you clarify?\
> => We set these hyperparameters to 1 and -1 and have added this information to the appendix. We did not explore other values.
> \
> &nbsp;
>
> (R4) At a more general level ...  such decisions is beyond the current state of art?\
> => There are current techniques that can make these decisions automatically—we fix them to focus specifically on the design of good transition policies. Because transitioning between actions is a problem that arises in many HRL approaches, we think that the problem of designing good transition policies that do not require retraining the subtask policies has wide relevance.

---

> > ### Comment · Reviewer_1ZCq · 2021-11-28
> > **Thank you for your response**
> >
> > Thank you for your answers to my questions. Based on my own appreciation of it and on the other reviewers' feedback, I hope your paper will get accepted.

---

### Official Review · Reviewer_M3VJ · 2021-11-02

**Correctness:** 4
**Technical Novelty And Significance:** 3
**Empirical Novelty And Significance:** 3
**Recommendation:** 6
**Confidence:** 2

**Main Review:**

Strengths
- The paper is well-written. I am not an expert in the area of HRL, but could still nicely follow the story of the paper, and the implementation details
- The problem studied is interesting, and seems challenging.
- The approach is well-justified

Weaknesses
- Some of the details in the background section slightly break the story and could be removed/moved to appendix, especially all the details about deep q-learning
- The number of baselines is quite marginal. I am not expert in the area and hence am unsure if more baselines could be added.


**Summary Of The Paper:**

The paper tackles the problem of connecting pre-trained policies $\pi_a, \pi_b$ in order to solve more complex tasks through abstraction. In order to do so, they leverage inverse RL (leveraging adversarial learning) to train a transition policy $\pi_{ab}$ aiming at transitioning between the two policies $\pi_a, \pi_b$. It is trained by enforcing its state-action occupancy to match that of the next pre-trained policy $\pi_b$ via inverse RL. They also propose leveraging a DQN with simple reward structure to control the transitioning from the trained transition policy $\pi_{ab}$ to $\pi_b$.

Contributions:
 - A relatively simple approach to transitioning between pre-trained policies, which has potential for impact when aiming to tackle complex tasks.
 - Empirical demonstration that their approach succeeds at tackling complex tasks by combining pre-trained policies successfully, and outperforming a single policy trained with PPO/SAC


**Summary Of The Review:**

I am not expert in the area of HRL, and hence I am only trying to provide an educated guess. I am recommending weak acceptance given the problem is interesting and the approach seems to make sense, along with an interesting set of experiments.

---

> ### Author Response · Authors · 2021-11-14
> **Author response for reviewer M3VJ**
>
> Thank you for your review.
>
> (W1) Some of the details in the background section ... about deep q-learning\
> => We have removed some details from the DQN section to improve the flow.
> \
> &nbsp;
>
> (W2) The number of baselines is quite ... if more baselines could be added.\
> => As far as we know, Lee et al. is the only other paper that studies the problem of constructing transition policies without retraining the subtask policies. We have attempted to make the comparison to their work as comprehensive as possible—we use the same experimental environments, we reproduce their hyperparameter settings, etc.

---

### Official Review · Reviewer_CNo5 · 2021-11-04

**Correctness:** 4
**Technical Novelty And Significance:** 3
**Empirical Novelty And Significance:** 2
**Recommendation:** 6
**Confidence:** 3

**Main Review:**

Strengths:

- the paper is clearly written (for comments on relatively minor
  exceptions of this, see below, all easy to fix)

- no unnecessary complexity of the description or the approach: idea
  and solution appear to do what is necessary for solution but not
  introduce complexity for complexity sake. More sophisticated
  solutions may be possible, but any extension or modification of the
  idea could then be benchmarked against the presented approach.

- I like the idea of training a transition policy for existing tasks

Weaknesses:

- The idea of learning the transition only after the first task is
  completed is an important limitation of the approach, dependent on
  the specific tasks. For some task combinations, it will be important
  to also modify the final steps of the first task, dependent on what the
  second task is supposed to do (or on how the second task begins).

  As an example, a robot with task A to approach an object, and then
  manipulate the object with task B might need to approach the object
  differently dependent on task B, or on the environment prior to
  beginning B. The transition policies here will only start after
  completing A, but I would suspect could also lead to backtracking,
  or oscillating behaviours.

  It would be good to see a discussion on how such issues might be
  addressed or avoided even if the approach in the current form
  doesn't do that.

- I was not sure if the experiments are the best to demonstrate or
  explore strength and weaknesses of the solution. I would be curious
  about a setup where the difficulty in transitions can be
  systematically changed. This would allow to also compare the
  adaptability of different methods, and a more thorough
  evaluation. Taking the example from above - an environment where the
  agent has to approach an object and move the object into a specific
  direction (eg by pushing towards a target state). This task is easy
  if agent, object, and target are aligned, but more difficult if the
  agent approaches the object from the side, and hard if the agent
  approaches the object from the target direction.


Other comments:

- Alg. 2: is the transition interval input to alg 2, or how is the
  condition in line 5 checked?
- Alg. 2: "run $q_{a,b}$" - I assume this one output (action:
  stay/switch), or is it two (Sec 3.2, first paragraph), and if it's
  two, why/what is it?
- Alg. 2: the condition in line 16 is ambiguous - should it run while
  $\pi_b$ has failed or not failed?
- it appears the assumption is the environment is reasonable static so
  that executing policies a and b can be scheduled, or at least
  policies implicitly check that they can still be executed at the
  beginning (there is no explicit option for replanning after $\pi_{a,b}$ terminates).
- the final sentence in the reproducibility statement is missing a verb


**Summary Of The Paper:**

The paper addresses the problem of learning to subsequently execute
tasks, and transitioning from a first task to a second task. The work
contributes a method to train transition policies, and a method to
decide when to start executing the second policy, stopping the
transition policy. The approach is evaluated on a few simulated robot
locomotion/manipulation tasks.


**Summary Of The Review:**

The paper presents a nice idea for an interesting problem, and a
strength of the paper is the level of complxity used to solve and
describe the problem. For complex combinations of tasks (where tasks
depend on each other), the approach may not be sophisticated enough as
the sole solution to the problem. Experiments and results are OK, but
could be more systematic.

---

> ### Author Response · Authors · 2021-11-14
> **Author feed for reviewer CNo5**
>
> Thank you for your review.
>
> (W1) The idea of learning the transition ...  or oscillating behaviours. \
> => We have updated the paper to discuss weakness #1 (in related work and in the conclusion).
> \
> &nbsp;
>
> (W2) I was not sure if the experiments  ... from the target direction. \
> => we agree that the creation of new tasks will be important for further advancing research in this area. We chose to use the exact same environments as Lee et al. to allow for direct comparison to their results, and we believe that the results on the locomotion tasks demonstrate that our approach handles challenging transitions substantially better.
> \
> &nbsp;
>
> (C1)  Alg. 2: is the transition interval input to alg 2, or how is the condition in line 5 checked \
> => We have added the transition interval to the inputs for Alg. 1 and 2.
> \
> &nbsp;
>
> (C2) Alg. 2: "run " $q_{a,b}$ ... and if it's two, why/what is it? \
> => One output—updated in the paper.
> \
> &nbsp;
>
> (C3) Alg. 2: the condition in line 16 is ambiguous ...  has failed or not failed?\
> => Should be “and” and not “or”—fixed.
> \
> &nbsp;
>
> (C4) it appears the assumption ...  can still be executed at the beginning \
> => Yes, we made this assumption and have made it explicit in a footnote on pg. 4. We think that it could be relaxed and we point that out.
> \
> &nbsp;
>
> (C5) the final sentence in the reproducibility statement is missing a verb\
> => We have added a verb.

---

### Decision · Program_Chairs · 2022-01-20

**Decision:**

Accept (Poster)

**Comment:**

Description of paper content:

The paper proposes a strategy to train a “transition policy” that can connect two pre-trained policies. The transition policy tries to reach state-action pairs that are within the occupancy distribution of the second policy using Inverse RL. The technique was evaluated on robot manipulation and locomotion problems.

Summary of paper discussion:

The discussion was not lengthy. The reviewers felt the paper was quite well-written, instructive, and novel, yet also implied the experimental results were less systematic than might be desired. All reviewers were weakly supportive of publication and made few critical comments. The salient ones concerned the experimental domains, the number of baselines, and the question of the generality of the approach.